# Protein Recovery from Rapeseed Press Cake: Varietal and Processing Condition Effects on Yield, Emulsifying Capacity and Antioxidant Activity of the Protein Rich Extract

**DOI:** 10.3390/foods8120627

**Published:** 2019-12-01

**Authors:** Karolina Östbring, Cecilia Tullberg, Stina Burri, Emma Malmqvist, Marilyn Rayner

**Affiliations:** 1Department of Food Technology Engineering and Nutrition, Faculty of Engineering, Lund University, P.O. Box 124, 221 00 Lund, Swedenemmamalmqvist@gmail.com (E.M.); marilyn.rayner@food.lth.se (M.R.); 2Department of Chemistry, Faculty of Engineering, Lund University, P.O. Box 124, 221 00 Lund, Sweden; cecilia.tullberg@biotek.lu.se

**Keywords:** rapeseed protein, by-product, protein recovery yield, emulsifying properties, oxidative effect

## Abstract

Protein was recovered from five varieties and a mixed blend of cold-pressed rapeseed press cake by leaching and precipitation in a water-based process, and the protein recovery yield varied from 26–41% depending on variety. Exposure for heat during protein recovery severely reduced the rapeseed proteins’ ability to stabilize the oil–water interface of emulsion droplets. Protein extract from Lyside had the best emulsifying properties of the varieties investigated. Oxidation rate was assessed by the Thiobarbituric Acid Reactive Substances (TBARS) method and rapeseed protein extracts from Epure and Festivo had higher capacity to delay oxidation compared with soy lecithin. There are possibilities to broaden the use of rapeseed whereby recovered rapeseed protein can be used as a plant-based multifunctional ingredient with emulsifying capacity and which has a delaying effect on oxidation.

## 1. Introduction

Rapeseed (*Brassica napus*, *Brassica rapa* and *Brassica juncea* rapeseed quality) is cultivated primarily for its oil content and fatty acid composition, but the process of extracting the rapeseed oil results in a currently underutilized by-product of a protein-rich meal or press cake. Finding ways to make use of this by-product is economically and environmentally important, since rapeseed is the third most cultivated oil crop globally (after palm oil and soy). The seeds contain 42% oil which is widely used as edible oil. The defatted meal or press cake contains up to 40% protein depending on the pressing process: Cold-pressed press cake contains 28–31% protein and hot-pressed meal contains 38–45% protein. Rapeseed protein has a well-balanced amino acid composition with higher amounts of S-amino acids than many other plants and can be considered a promising source of food protein [1]. However, the utilization of rapeseed protein is limited due to the presence of glucosinolates, phytates and phenolics [2], and the rapeseed protein is therefore used as feed for dairy cattle, poultry and pigs, as well as limited use in aquaculture [3]. Removal of anti-nutritional factors is crucial for allowing the rapeseed protein to be used as human food. Several attempts have been made, and the most promising approach is the use of filtration techniques [4].

In addition to the nutritional value of the rapeseed by-product, the proteins have also been reported to have an emulsifying and anti-oxidative capacity, which implies that they potentially can be used as a functional ingredient in the food industry [5,6,7,8]. Rapeseeds contain both storage proteins (cruciferin and napin), which account for 80–90% of the total protein content, and oil body proteins (oleosin and caleosin), accounting for the rest [9]. In oilseeds, the triacylglycerols are stored as oil bodies surrounded by both a monolayer of phospholipids as well as oil body proteins to stabilize the oil–water interface. Oil bodies occur as individual entities and do not congregate even when subjected to extreme environmental stress, such as moisture or temperature fluctuations [5]. The stability of oil bodies has been attributed to the oleosin protein structure. The *N*-terminal is folded into both a hydrophilic and a hydrophobic region arranged parallel to each other, with the hydrophilic region facing the aqueous phase and the hydrophobic region facing the lipid phase. The oil body proteins also have a long double-bonded hydrophobic region and the folding allows a 180 °C turn, anchoring the protein to the oil droplet. This hydrophobic region is the longest found in proteins from organisms. The *C*-terminal end of the protein forms an alpha helix at the surface of the phospholipid layer [10]. By this structure, the rapeseed proteins (particularly oleosin) can act as emulsifying agents in food emulsions [11].

Many of our most beloved foods are in emulsion form, such as ice cream, beverages, sauces and salad dressings. An emulsion is a mixture of two immiscible liquids, often water and oil, where one phase is dispersed in the other in the form of small droplets. Emulsifiers are needed to stabilize the oil–water interface to prevent coalescence. Emulsifiers can be either synthetic or natural and the most common groups of natural emulsifiers are proteins and complex carbohydrates [12]. Some synthetic emulsifiers (Carboxymethylcellulose and Polysorbate 80) have been reported to impair the gastrointestinal integrity in rats and cause severe damage to the intestinal wall, with translocation of bacteria to the blood as a consequence [13]. An interest in alternative natural emulsifiers is hence arising in the food industry.

A vexing problem within the food industry is the occurrence of lipid oxidation, a chemical reaction causing food deterioration due to unpleasant changes in aroma, colour and texture. If lipid oxidation can be delayed in lipid-rich foods, the shelf life of the products can be prolonged. An increasing degree of unsaturation of the fatty acids has been connected to an increasing level of lipid oxidation [14], and flaxseed oil with its high level of α-linolenic acid is hence an oil which is prone to lipid oxidation. There are several studies where protein isolates used as emulsifying agents have been found to act protectively against lipid oxidation [15,16,17]. Furthermore, various peptide hydrolysates from rapeseed proteins have been found to possess promising anti-oxidative features [18,19,20]. However, the oxidative effect of whole rapeseed protein isolates used as emulsifier has to our knowledge not yet been investigated.

Rapeseeds can either be hot-pressed or cold-pressed to facilitate the liberation of oil from the seeds. The vast majority of the global production of rapeseed oil is hot-pressed due to higher oil yields at higher temperatures and typically solvents are added to the rapeseed meals to further increase the oil yield. There are several studies on hot-pressed or solvent-extracted rapeseed meal and its functional properties [21,22], and it has been reported that both heat and solvents reduce the emulsifying properties of rapeseed proteins [23]. However, only a limited number of studies have used cold-pressed rapeseed press cake as starting material [24], and fewer still have used industrial cold-pressed side streams rather than material processed in the lab. In the case of hot-pressed rapeseed protein, it has also been demonstrated that botanical variety influences both protein recovery yield and functional properties [21], but to the authors’ knowledge no such studies have investigated different botanical varieties of industrially cold-pressed starting material.

The objectives of the present study were to investigate protein recovery yield, emulsifying properties and oxidative stability of five industrially cold-pressed Scandinavian rapeseed varieties and a mixed blend.

## 2. Materials and Methods

### 2.1. Materials and Chemicals

The rapeseed varieties investigated in the present study were Alegria, Epure, Festivo, Lyside, V316OL and a mixed blend of different varieties (Alegria, Epure and a few unknown varieties). Lyside is spring oilseed rape whereas the other varieties are winter oilseed rape varieties. The different varieties were cultivated at the same farm and harvested in southern Sweden in 2016 (latitude N 55°29.42.697′, longitude E 14°11.0.367′). Cold-pressed rapeseed press cake (*Brassica napus*) of different varieties and cold-pressed flaxseed oil was a kind gift from Gunnarshögs Jordbruk AB, Hammenhög, Sweden. All rapeseed varieties were cold-pressed separately at Gunnarshögs Jordbruk AB without the use of solvents and the temperature during oil extraction was not exceeding 35 °C. The rapeseed press cake was stored in freezer (−18 °C) until the start of the recovery processes (described in Section 2.2). The cold-pressed flaxseed oil was pressed in the same manner as the rapeseed the same day as the onset of the storage experiments in Section 2.5. The flaxseed oil was kept at 4 °C in the dark for three hours until the onset of the experiments.

De-oiled soy lecithin (Epikuron 100 P IP) was purchased from Cargill (Minneapolis, MN, USA). Bovine serum albumin (BSA) (A-3059), citric Acid (C_6_H_8_O_7_, CAS 77-92-9), sodium chloride (NaCl, CAS 7647-14-5), sodium dihydrogen phosphate monohydrate (H_2_Na PO_4_·H_2_O, CAS 7558-80-7), di-sodium hydrogen phosphate dodecahydrate (Na_2_HPO_4_·12H_2_O, CAS 10039-32-4), trichloroacetic acid (TCA, 76-03-9), and sodium hydroxide (NaOH, CAS 1310-73-2) were purchased from Merck (Darmstadt, Germany). 1,1,3,3-tetramethoxypropane (CAS 102-52-3), 2-thiobarbituric acid (98%) (TBA, CAS504-17-6) were purchased from Sigma-Aldrich (St. Louis, Kansas, MO, USA). Miglyol 812 was purchased from Sasol AG, Germany. Hydrochloric acid (HCl, CAS 7647-01-0) was purchased from VWR Chemicals (Fontenay-sous-Bois, France). All other chemicals were of analytical grade.

### 2.2. Protein Recovery from Rapeseed Press Cake

Protein was recovered from rapeseed press cake (RSPC) from the different varieties and the mixed blend (see Figure 1 for the schematic process). The protein recovery procedure was identical for all raw materials. The procedure was a modified version of the procedure previously described by Wijesundera [25].

Cold-pressed RSPC (50 g) was ground in a knife mill (Grindomix GM 200, Retsch, Germany) at 2500 rpm for 20 s. The press cake was leached in basic solution (1:10 *w/w*) with pH 10.5. The dispersion was mixed for 10 min (IKA Labortechnik, Eurostar digital) at 160 rpm, pH was re-adjusted to 10.5, and the dispersion was mixed at 4 °C for 3 h. After incubation, the dispersion was centrifuged (Beckman Coulter, Allegra^®^ X-15R Centrifuge, Brea, CA, USA) for 30 min at 5000× *g* in 4 °C. The supernatant was collected, and pH was adjusted to 5.0 with citric acid powder. One half of the experimental set was heat-treated at 80 °C on an induction stove with a coming up time of approximately 4 min and a holding time of 3 s, followed by immediate cooling in a cold-water bath until ambient temperature. The other half of the experimental set was not subjected to heat treatment. The dispersion was centrifuged (Beckman Coulter, Allegra^®^ X-15R Centrifuge, Brea, CA, USA) for 30 min at 5000× *g* in 4 °C. The precipitates were collected and were used in wet form for all other experiments. The recovery process was performed in triplicate for each variety and for the mixed blend.

### 2.3. Protein Quantification and Dry Matter Analysis

Protein content was analysed by the Dumas method for both the rapeseed press cake from each raw material as well as for the subsequent protein precipitate. Nitrogen content was determined by the elemental analyser Flash EA 1112 (Thermo Electron Co., Waltham, MA, USA) blanked with air, with aspartic acid as reference. Approximately 25 mg material was placed in a tin cylinder (diameter 30 mm) for analysis. A conversion factor of 6.25 was used to calculate protein content. Each batch was analysed in triplicates (nine measurements per variety/processing method). The protein recovery yield was calculated by Equation (1):(1)$$Protein\;recovery\;yield\;\left(\%\right)=\frac{\mathrm{Protein}\;\mathrm{amount}\;\mathrm{in}\;\mathrm{rapeseed}\;\mathrm{protein}\;\mathrm{precipitate}\;\left(g\right)}{\mathrm{Protein}\;\mathrm{amount}\;\mathrm{in}\;\mathrm{press}\;\mathrm{cake}\;\left(g\right)\;}\;\times \;100\%.$$

Dry matter content was determined according to the official method of analysis (AOAC 2007). The temperature was adjusted to 102 °C and analyses were performed in triplicate.

### 2.4. Preparation of Emulsions

Oil-in-water emulsions were prepared in triplicates in glass test tubes with 2 mL continuous phase (0.005 M phosphate buffer, 0.2 M NaCl, pH 7), 1 mL dispersed phase (Miglyol 812, Sassol AG, Germany) and a varying amount of protein extract. Concentrations of 2, 4, 8, 16 and 32 mg rapeseed protein/mL oil of each rapeseed protein precipitate were used. The emulsions were homogenized with a mixer (Ystral D-79282, Ballrechten- Dottingen, Germany) at 22,000 rpm for 60 s. The emulsions were thereafter stored for 1 h in 4 °C prior to particle size analysis. Emulsions were made in triplicates.

### 2.5. Particle Size Distribution of Emulsions

Particle size distribution of emulsions with rapeseed protein as emulsifier were analysed with a laser diffraction particle analyser (Mastersizer 2000 Ver 5.60, Malvern, Worcestershire, UK). The pump velocity was 2000 rpm with 100 mL MilliQ^®^ water in the sampling chamber. Each emulsion replicate was measured three times and the average was reported. The refractive index (RI) was 1.45 for the miglyol oil and 1.33 for the water. The obscuration was between 10–20%. From the particle size distribution, *d*_32_, *d*_43_, mode, span and volume (%) were calculated. Emulsion droplets in an emulsion have different sizes and constitute a particle size distribution. From this distribution different particle sizes can be calculated, such as the surface weighted mean (*d*_32_) and the volume weighted mean (*d*_43_). Mode is the top of the peak and span is the width of the peak in the volume distribution.

To investigate the emulsion stability over time, emulsions were prepared as described in Section 2.4, with 8 mg rapeseed protein/mL oil from different varieties, soy lecithin or BSA as emulsifier, phosphate buffer as the continuous phase and flaxseed oil as dispersed phase. The rapeseed proteins used in the emulsion stability test was all heat-treated at 80 °C for a few seconds during the protein extraction process (Figure 1) in order to provide a fair comparison to the controls (soy lecithin and BSA), as both had been exposed to mild heat treatment during spray drying to a powder. The chosen protein concentration was the concentration after which a plateau was reached and no further reduction in particle size was found regardless of protein concentration. The emulsions were incubated in the dark at 30 °C to accelerate destabilization processes. Particle size distribution was analysed as described above after 0, 8 and 34 days of incubation. Emulsions were prepared in duplicates and each emulsion was measured twice.

### 2.6. Rate of Lipid Oxidation

The rate of lipid oxidation was investigated for emulsions using rapeseed protein precipitates from the different varieties, the mixed blend as well as soy lecithin and BSA as emulsifiers. The rapeseed protein extracts used were all heat-treated (Figure 1). To investigate possible differences between the different varieties, the rapeseed proteins’ capacity to delay lipid oxidation was quantified using Thiobarbituric acid reactive substances (TBARS) in an emulsion model incubated in accelerated conditions. Emulsions were prepared in glass tubes with 2 mL phosphate buffer, 1 mL cold-pressed flaxseed oil and 32 mg rapeseed protein by mixing (Ystral D-79282, Ballrechten- Dottingen, Germany) at 22,000 rpm for 60 s. For comparison, emulsions with soy lecithin and BSA as emulsifiers were prepared. The concentration of emulsifiers was scaled to match the particle size distribution of emulsions with rapeseed protein as emulsifier. Soy lecithin (2.75 mg) and BSA (0.75 mg) were pre-mixed with phosphate buffer using a magnetic stirrer (Ystral D-79282, Ballrechten- Dottingen, Germany, 22,000 rpm, 60 s) and were thereafter combined with the other components. The emulsions were prepared as above and were incubated in the dark at 30 °C for 0, 1, 2, 6, 8, 16 and 34 days. All emulsions were prepared in duplicates.

After incubation the emulsions were mixed (Ystral D-79282, Ballrechten-Dottingen, Germany, 22,000 rpm, 10 s) to allow representative sampling and 0.5 mL of the emulsions were rapidly transferred to plastic tubes, to which 2.5 mL TBA-solution (0.375% (*w/v*) TBA, 15% (*w/v*) TCA and 0.25M HCl was added. The samples were again mixed for 10 s. The tubes were heated in a water bath at 90 °C for 10 min and were thereafter rapidly cooled under cold running water until ambient temperature. The samples were centrifuged (Beckman Coulter, Allegra^®^ X-15R Centrifuge, Brea, CA, USA) for 30 min, 5000× *g*, 25 °C. The absorbance of the supernatant was measured at 532 nm with a spectrophotometer (Varian Inc., Vary 50 UV-Vis, Palo Alto, CA, USA) against a blank TBA-solution. Furthermore, the absorbance at 600 nm was subtracted to remove the impact of turbidity. 1,1,3,3-tetramethoxypropane in a range of 1–10 ppm was used as standard (*r*^2^ = 0.9958, *y* = 0.181492*x* + 0.023390) and the concentration of the oxidation product malondialdehyde (MDA) could be calculated. The absorbance was quantified twice for each emulsion.

### 2.7. Statistical Analysis

Area under the curve (AUC) was calculated for *d*_43_, *d*_32_, mode, span, *d*_43_ after storage and the concentration of MDA in oxidation trials by numerical integration using the trapezoidal rule method. The emulsifying properties, emulsion stability over time and oxidative effect for each rapeseed protein precipitate or emulsifier could thereby be quantified.

Data were analysed using SPSS software version 22 (IBM). Univariate general linear model with Tukey’s test was performed to investigate significant differences in parametric data sets (dry matter, protein concentration on dry basis, protein recovery yield and *d*_32_). A Kruskal–Wallis test with Bonferroni-adjusted pairwise comparison was performed to investigate any significant difference in non-parametric data sets (mass of sediment, *d*_43 _at 8 mg protein/mL oil, AUC for *d*_43_, *d*_32_, mode, span, *d*_43_ after storage and oxidation trials). Differences were considered significant if *p* < 0.05.

## 3. Results and Discussion

### 3.1. Effect of Rapeseed Variety on Protein Yield

Proteins were recovered and analysed from cold-pressed rapeseed press cake (RSPC) from five different varieties and a mixed blend and the recovery yield was dependent on variety (Table 1). For proteins recovered without heat, the rapeseed varieties Alegria and Epure had the highest protein yield (40% ± 1%, 41% ± 1%) and a mixed blend of RSPC had the lowest yield (29% ± 2%). It should be mentioned that this study was performed with rapeseed varieties from one single year. Although there were distinctive differences in protein recovering yield among the investigated varieties, further studies including rapeseed varieties from several years should be performed to present more robust data.

Common strategies to separate proteins in dispersions include the addition of acid and/or heat. Adjustment of pH to the proteins’ isoelectric point induce precipitation of the proteins with association and sedimentation as consequence. Heat is used to denature the proteins, which will promote aggregation and sedimentation as well. In the present study exposure to heat (80 °C in a few seconds) did not improve the protein recovery yield from the varieties investigated (Table 1). The mixed blend of RSPC was a blend of Alegria and Epure but also contained a few unknown varieties. Since Alegria and Epure both had a high protein recovery yield, the unknown varieties included in the mix should have had significantly lower protein yield, which reduced the average yield for this specific sample.

The protein recovery yields in the present study were higher for specific rapeseed varieties compared with previously reported values [21,26] using similar recovery processes and starting material. In the present study, 40–41% of the protein could be collected in the precipitate from the varieties Alegria and Epure, whereas Rommi et al. reported a protein recovery of 28% [26]. In the present study, a slightly higher pH was used (pH 10.5 instead of 10.0) and the dispersion was incubated for a substantially longer time: four hours instead of one hour in the study by Rommi et al. The longer incubation time together with the slightly higher alkali pH may be reasons for the higher recovery yield in the present study. Ghodsvali et al. used a hot-pressed rapeseed meal treated with hexane and reported that 12–15% of the proteins could be collected in the precipitate. Both heat and solvents as hexane have been reported to reduce the rapeseed protein recovery due to denaturation [27].

The mass of sediment collected was dramatically increased when heat was applied in the protein recovery process compared with the non-heat treatment, but the dry matter content and the protein concentration was reduced (Table 1). One reason for the increased mass of protein precipitates may be that proteins are known to form gels when subjected to heat treatment, hence the increased capacity to hold water [28]. Hermansson et al. found that the gel-forming capacity for soy proteins was increased with temperature, and for soy protein dispersed in water a steep increase in shear stress of the gel could be seen already at 70 °C [29]. The phenomenon was attributed to dissociation of the subunits of the globular proteins followed by an aggregation process, and the temperature for onset of gelation was close to the denaturation temperature for soy protein. The temperature for onset of gel formation may be similar for rapeseed protein, although not investigated in the present study.

Heat treatment in the recovery process had a significant effect on the protein concentration on dry basis. The protein concentration in the precipitates for samples recovered without heat ranged between 57% and 70%, whereas when heat was applied, the protein concentration was reduced to 41–54% on a dry basis (Table 1). Temperature affected the mixed blend most dramatically of the studied varieties. Without heat in the recovery process, the protein concentration was 70% but when heat was applied the protein concentration was reduced to 41%. Other varieties such as Epure did not show such large temperature dependence; the difference in protein concentration only ranged from 62% when no heat was applied to 55% when heat was applied. The samples exposed to heat during the extraction process adsorbed more water, hence having a higher total mass but lower protein concentration. One hypothesis is that the heat-treated proteins together with the acid created a weak gel with higher water holding capacity compared with non-heat-treated proteins [28].

All varieties investigated have the same price when purchased as seeds, and Alegria, Epure and Festivo have a similar oil content and crop yield. Lyside is a white flowering variety with a flavour profile differing from the other varieties, and the oil is sold for an approximately 30% higher price. However, Lyside has the lowest crop yield of the varieties investigated—an approximately 25% lower crop yield than the other varieties—and due to the low crop yield and low protein yield, Lyside was concluded to not being an optimal variety for plant protein recovery. V316OL is a rapeseed variety with HOLL profile (High Oil, Low Linolenic acid) used for deep frying food applications. For protein recovery in bulk, the present study indicates that the varieties Alegria or Epure should be used due to high crop yield and high protein recovery yields, which will be economically favourable.

### 3.2. Emulsifying Properties

Emulsion with rapeseed protein extracts displayed decreased droplet sizes with an increased amount of protein in the formulation. The trend was similar for all rapeseed varieties tested within each experimental set (heat treatment or no heat treatment) (Table 2). The trend with decreased emulsion droplet size with increased amount of protein was similar for both the volume weighted mean (*d*_43_), surface weighted mean (*d*_32_) and mode (Table 2), indicating an even droplet size distribution with no substantial accumulation of neither smaller nor larger droplets. When area under the curve was calculated to include all protein concentrations investigated, there were no statistical differences between the rapeseed varieties investigated for *d*_43_, mode and span. For *d*_32_, the protein extract from Alegria had significantly better emulsifying capacity compared with V316OL, indicating an increased number of small droplets in the particle size distribution for Alegria. Heat in the protein recovery process severely reduced the emulsifying capacity for varieties investigated for both *d*_43_, *d*_32_ and mode. For span, which is the width of the peak in the particle size distribution, all varieties except Epure had a significantly wider peak when no heat was applied in the recovery process compared with when heat treatment was used.

When the protein concentration in emulsions increases, a larger number of protein molecules associates with the oil–water interface and, at some level, the surface become saturated. In the present emulsion experiment, the critical concentration was 8 mg protein/mL oil. At concentrations above this critical concentration, the particle size of emulsion droplets was not further reduced, and emulsion droplet size converged for emulsions with the different rapeseed varieties (Figure 2).

In the absence of heat in the protein recovery process, the droplet diameter *d*_43_ at 8 mg protein/mL oil ranged between 42–56 μm (average 48 μm) depending on the rapeseed variety (Figure 2). Application of heat in the protein recovery process had a pronounced effect on the proteins’ emulsifying capacity and the mean droplet diameter increased to a range of 81–98 μm (average 91 μm for *d*_43_ at 8 mg protein/mL oil) (Figure 2).

Protein extracted from Lyside stabilized the oil–water interface more effectively compared to protein extracts from the mixed one, with smaller emulsion droplets as a consequence when no heat was applied (solid bars in Figure 2). The other varieties had a similar emulsifying capacity as both Lyside and Mixed. When heat was applied, proteins from Mixed and Alegria was more efficient in stabilizing the oil–water interface compared to Epure, whereas the other varieties did not differ significantly (brick pattern in Figure 2).

Particle size distributions were bimodal with a small peak around 10 μm, representing protein aggregates, and a large peak around 40 μm (Figure 3a) and 100 μm (Figure 3b), representing the emulsion droplets. The particle size distributions were relatively narrow with a span of about 1.

The reported average droplet size (*d*_43_) was smaller compared to values reported by Tan et al. [30], a study in which emulsions were produced with an acid-precipitated isolate of cold-pressed canola as emulsifier (these emulsions had a droplet size of *d*_43_ around 60 μm). The difference between the present study and the study reported by Tan et al. might be due to the use of a different acidic pH. For the present study, a pH of 5.0 was used whereas Tan et al. used a pH of 4.0, and proteins recovered at pH 5.0 might have a slightly higher emulsifying capacity. Oleosin, an oil body protein in rapeseed, is known to have high emulsifying capacity due to its unique structure [10]. Oleosin has an isoelectric point (IP) of 6.5 and the precipitation pH in the present study (pH 5) is closer to oleosins’ IP compared to the pH used in the study reported by Tan et al. The rather low emulsifying capacity could be due to high molecular weight polypeptides (<250 kDa) remaining in the isolate [30]. In addition, the presence of disulphide bonds in canola protein isolates could further reduce the overall flexibility of the high molecular weight polypeptide giving the lower emulsifying capacity [30].

Aluko et al. found that there were significant differences in the emulsifying capacity of acid-precipitated protein isolates from different botanical varieties of *Brassica napus* [31]. The authors argued that some varieties might contain non-protein substances, which interfered with emulsion formation. The present paper investigated varieties specific for Sweden, whereas Aluko et al. studied rapeseed varieties from Canada. There seem to be some overall variation in protein composition and functional properties in different varieties of rapeseed. In the present study, Lyside had the best emulsifying capacity, although the majority of the investigated varieties in the present paper had similar emulsifying capacity.

### 3.3. Emulsion Stability

Emulsions were formulated with flaxseed oil as the lipid phase and either rapeseed protein precipitate, BSA or soy lecithin as emulsifiers to investigate emulsion stability and oxidation rate in accelerated conditions (30 °C for 34 days). The droplet size (*d*_43_) was increasing over time for all emulsions investigated (Figure 4). Emulsions with rapeseed proteins as emulsifiers followed the same trend and choice of rapeseed varieties did not affect the emulsion destabilization processes significantly. It should be mentioned that the emulsions with BSA as emulsifier underwent phase separation during the experiments (Figure 4a) and the particle sizes were exceeding 130 μm around Day 8, which is equal to the particle size auto-generated by the Malvern Mastersizer equipment when pure flaxseed oil is introduced to the dispersion unit. Emulsions with soy lecithin as emulsifier were more stable than the other emulsions in the early phase of the experiments, but after incubation for 34 days, the particle size was larger than for emulsions with rapeseed proteins (122 μm vs. 95–106 μm) as emulsifier, although not significantly. Protein extracted from rapeseed independent of variety as emulsifier produced emulsions with equal storage stability as soy lecithin, and significantly more stable emulsions compared to BSA.

### 3.4. Lipid Oxidation

Proteins recovered from Epure and Festivo delayed the lipid oxidation to a significantly greater extent compared with emulsions produced with soy lecithin as emulsifier (*p* < 0.05) (Figure 5).

It is well established that proteins generally have been found to protect the oil phase against oxidation [7] and the anti-oxidative effect of rapeseed protein has previously been reported [8]. The antioxidative effect of proteins in general have been attributed to the capacity to chelate metal ions or to act as antioxidants by inactivation of free radicals through interactions with hydrophobic amino acids [32]. Rapeseed protein has been reported to have a high proportion (about 40%) of hydrophobic amino acids [8]. The barrier properties of protein-stabilized interfaces where proteins form a relatively thick and viscoelastic interface around lipid droplets can also be important in terms of reducing oxidative damage. Proteins thereby inhibit oxidation through their physical accumulation at lipid–water interfaces, thus forming physical barriers between water-soluble pro-oxidants and lipids [33]. This barrier has been suggested to be at least partly responsible for the higher oxidative stability of emulsions with proteins as emulsifiers compared to emulsions with surfactants as emulsifiers [33,34,35].

## 4. Conclusions

Proteins were recovered from cold-pressed rapeseed press cake from different varieties. For protein recovery in bulk, the present study indicates that the varieties Alegria or Epure should be used due to high protein recovery yields, which will be economically favourable. The emulsifying capacity was severely reduced after exposure to heat during protein recovery, independent of variety, and when no heat was applied, protein extract from Lyside had the best emulsifying capacity. Rapeseed extract from all varieties investigated produced emulsions with similar stability against destabilization processes over time compared with soy lecithin, and a higher stability compared with BSA. Rapeseed protein extracts from Epure and Festivo had significantly higher capacity to delay oxidation compared with soy lecithin. There are possibilities to broaden the use of rapeseed, not only for producing edible vegetable oil, but also to formulate different foods and ingredients from the protein fraction in the press cake. Recovered rapeseed protein is promising candidates to be used as plant-based multifunctional ingredients with emulsifying capacity and a delaying effect on oxidation. This study should be considered preliminary and more studies are needed where rapeseed of the different varieties are followed over several years to conclude if the results from the present study are valid also over time.

## Figures and Tables

**Figure 1 foods-08-00627-f001:**
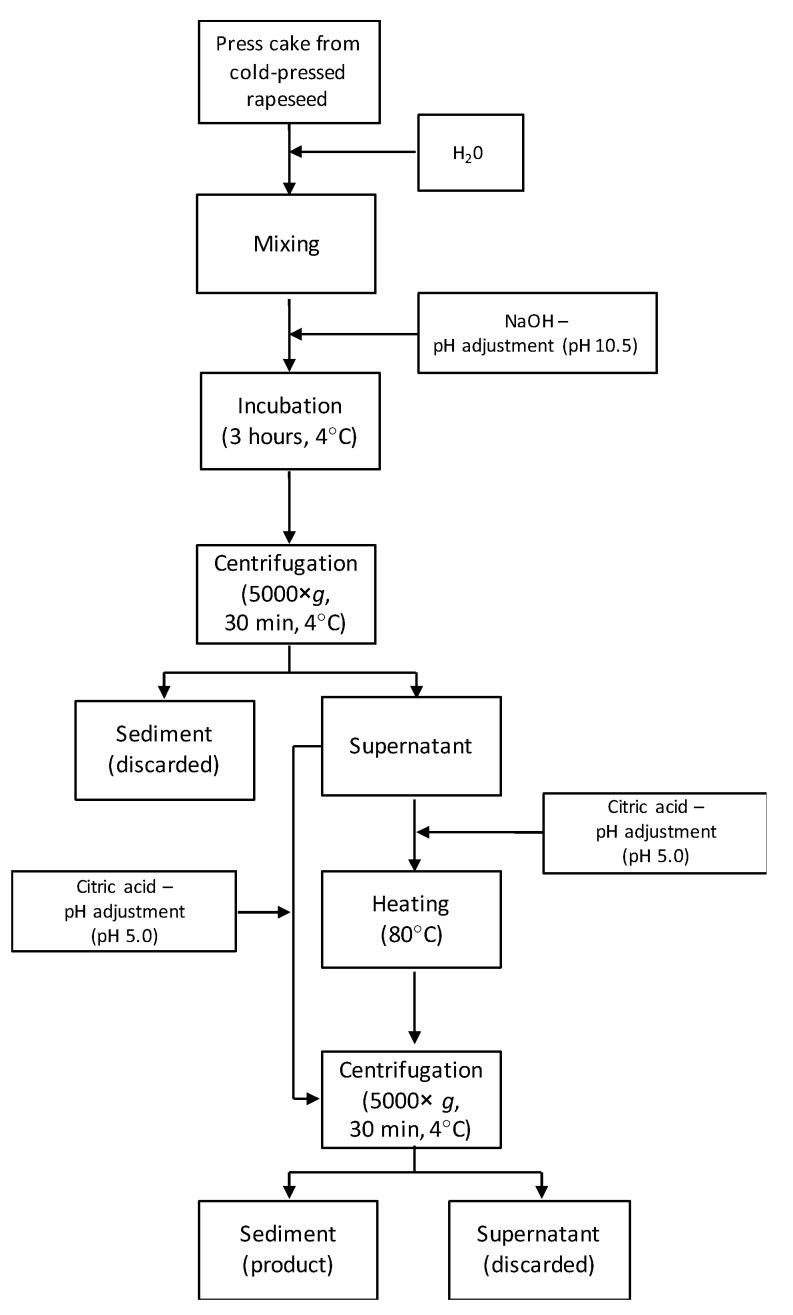
Schematic illustration of the extraction process of rapeseed protein from cold-pressed rapeseed press cake.

**Figure 2 foods-08-00627-f002:**
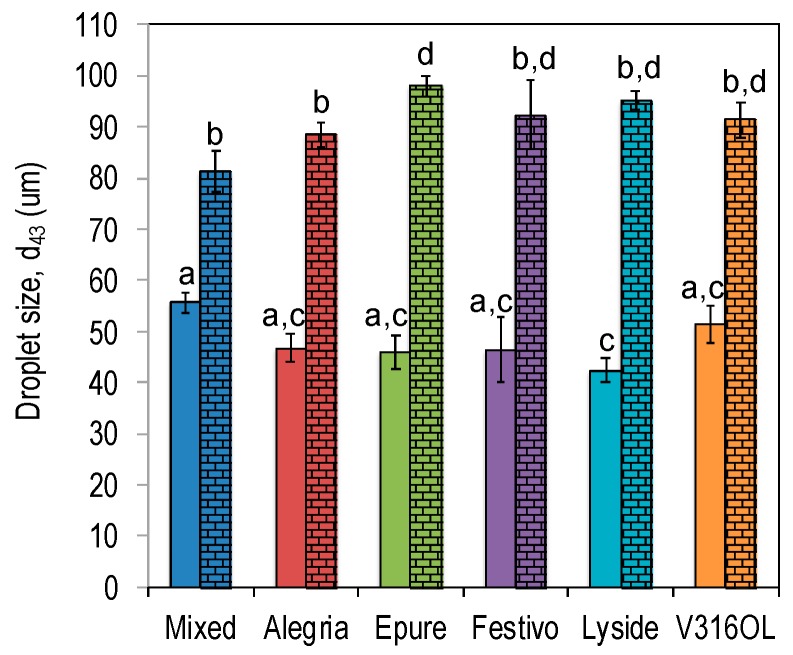
Droplet size (*d*_43_) for emulsions with protein precipitates from different rapeseed varieties as emulsifiers (8 mg/mL oil). Solid bars represent no heat in the recovery process and bars with a brick pattern represent heat included in the recovery process. Different letters indicate significant difference (*p* < 0.05).

**Figure 3 foods-08-00627-f003:**
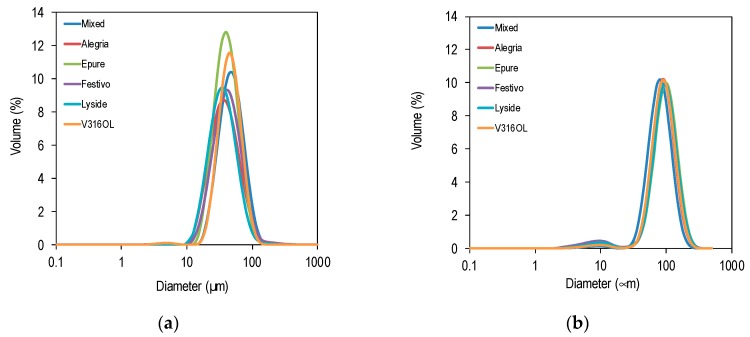
Size distribution of emulsion droplets stabilised by rapeseed protein precipitate from different varieties (8 mg protein/mL oil). (**a**) No heat treatment during the protein recovery process. (**b**) With heat treatment during the protein recovery process.

**Figure 4 foods-08-00627-f004:**
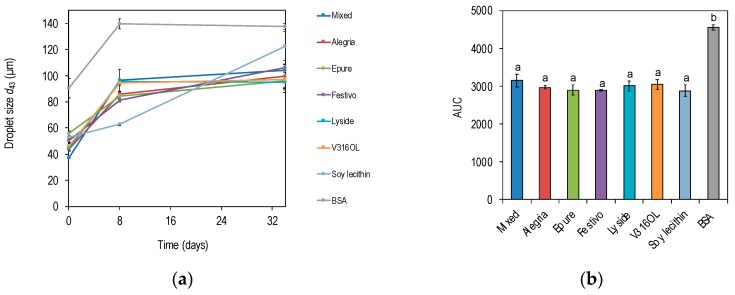
(**a**) Droplet size (*d*_43_) as a function of time for emulsions incubated in accelerated conditions (30 °C). (**b**) AUC for emulsion droplet size (*d*_43_) as a function of time. Emulsions were produced with rapeseed proteins from different varieties, soy lecithin or bovine serum albumin (BSA).

**Figure 5 foods-08-00627-f005:**
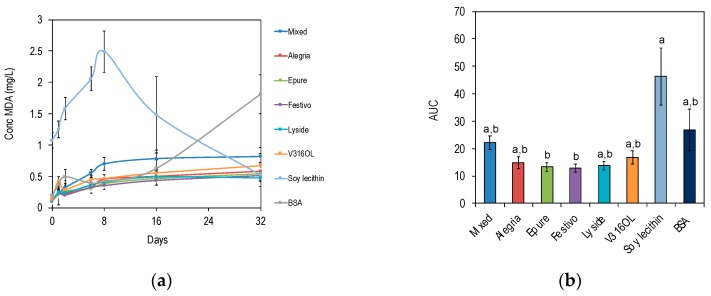
Emulsions with protein from different rapeseed varieties or commercial emulsifiers (bovine serum albumin (BSA), lecithin). The emulsions were incubated in accelerating conditions (30 °C) for 32 days. (**a**) Malondialdehyde (MDA) levels (mg/L) as a function of time. (**b**) AUC for MDA (mg/L) as a function of time. Different letters indicate significant difference (*p* < 0.05).

**Table 1 foods-08-00627-t001:** Physico-chemical properties and protein recovery yield of rapeseed protein precipitate from different varieties. Data are given as mean ± standard deviation. RSPC = rapeseed press cake. Different letters in columns indicate significant difference, *p* < 0.05).

Rapeseed Cultivar	Heat Treatment During Protein Recovery	Mass of Sediment (g)/100 g RSPC	Dry Matter (%)	Protein Concentration on Dry Basis (%)	Protein Recovery Yield (%)
Mixed	no	30 ± 2 ^a^	32 ± 0.4 ^a^	70 ± 3 ^a^	29 ± 2 ^a^
Alegria	no	43 ± 5 ^a,b,c^	33 ± 0.9 ^a,b^	64 ± 5 ^a,b,c^	40 ± 1 ^b^
Epure	no	45 ± 2 ^a,b,c^	35 ± 1 ^b^	62 ± 1 ^a,b,c^	41 ± 1 ^b^
Festivo	no	38 ± 3 ^a,b,c^	35 ± 1 ^b^	70 ± 2 ^a,b^	37 ± 2 ^c^
Lyside	no	48 ± 3 ^c^	33 ± 0.4 ^a,b^	57 ± 2 ^c^	36 ± 1 ^c^
V316OL	no	37 ± 2 ^a,b,c^	34 ± 0.3 ^b^	61 ± 0.5 ^b,c^	32 ± 2 ^c^
Mixed	yes	78 ± 8 ^d^	19 ± 0.6 ^c^	41 ± 2 ^d^	26 ± 1 ^a^
Alegria	yes	110 ± 2 ^d^	21 ± 0.9 ^c^	42 ± 5 ^d,e^	41 ± 4 ^b^
Epure	yes	91 ± 1 ^d^	20 ± 0.4 ^c^	54 ± 2 ^f^	41 ± 1 ^b^
Festivo	yes	84 ± 2 ^d^	19 ± 1 ^c^	51 ± 3 ^e,f^	33 ± 1 ^c^
Lyside	yes	100 ± 7 ^d^	21 ± 0.7 ^c^	42 ± 3 ^d,e^	34 ± 3 ^c^
V316OL	yes	83 ± 2 ^d^	19 ± 0.2 ^c^	50 ± 1 ^d,e,f^	33 ± 1 ^c^

**Table 2 foods-08-00627-t002:** (**a**) Droplet diameter, span and mode of emulsions with rapeseed protein precipitate (no heat) from different varieties as emulsifiers. Data are given as mean ± standard deviation. (**b**) Droplet diameter, span and mode of emulsions with rapeseed protein precipitate (with heat) from different varieties as emulsifiers. Data are given as mean ± standard deviation.

Rapeseed Variety	Protein Concentration (mg protein/mL oil)	*d*_43_ (μm)	*d*_32_ (μm)	Span	Mode (μm)
(**a**)					
Mixed	2	90 ± 11	81 ± 10	0.84 ± 0.1	85 ± 10
	4	67 ± 4	56 ± 7	1.1 ± 0.1	62 ± 4
	8	56 ± 2	45 ± 4	1.2 ± 0.1	50 ± 1
	16	45 ± 2	29 ± 1	1.4 ± 0.1	41 ± 1
	32	38 ± 3	23 ± 1	1.6 ± 0.1	36 ± 4
Alegria	2	86 ± 14	74 ± 19	1.0 ± 0.2	81 ± 15
	4	59 ± 2	48 ± 3	1.2 ± 0.1	52 ± 3
	8	47 ± 3	36 ± 3	1.4 ± 0.1	39 ± 3
	16	49 ± 13	32 ± 5	1.4 ± 0.2	44 ± 17
	32	43 ± 10	23 ±5	1.7 ± 0.1	38 ± 15
Epure	2	78 ± 8	68 ± 10	0.88 ± 0.2	75 ± 8
	4	57 ± 5	50 ± 5	0.92 ± 0.1	53 ± 4
	8	46 ± 3	39 ± 2	1.1 ± 0.1	42 ± 3
	16	40 ± 2	32 ± 2	1.2 ± 0.04	36 ± 1
	32	38 ± 1	26 ± 0.3	1.3 ± 0.1	36 ± 1
Festivo	2	68 ± 8	63 ± 6	0.96 ± 0.1	70 ± 5
	4	63 ± 14	53 ± 10	1.2 ± 0.2	62 ± 10
	8	46 ± 6	39 ± 5	1.3 ± 0.2	46 ± 4
	16	39 ± 6	28 ± 4	1.3 ± 0.1	37 ± 3
	32	36 ± 4	23 ± 1	1.6 ± 0.1	34 ± 2
Lyside	2	70 ± 5	57 ± 5	1.1 ± 0.2	66 ± 7
	4	53 ± 3	43 ± 6	1.2 ± 0.2	47 ± 5
	8	42 ± 2	34 ± 3	1.3 ± 0.2	36 ± 3
	16	38 ± 3	28 ± 3	1.4 ± 0.1	31 ± 3
	32	32 ± 3	22 ± 2	1.5 ± 0.1	27 ± 3
V316OL	2	80 ± 6	73 ± 5	0.8 ± 0.1	76 ± 5
	4	58 ± 5	51 ± 4	1.0 ± 0.1	54 ± 5
	8	51 ± 4	44 ± 4	1.0 ± 0.1	48 ± 3
	16	50 ± 1	43 ± 1	1.0 ± 0.1	47 ± 1
	32	48 ± 2	31 ± 1	1.0 ± 0.1	48 ± 2
(**b**)					
Mixed	1	140 ± 9	110 ± 10	1.0 ± 0.1	123 ± 8
	2	110 ± 6	96 ± 5	0.99 ± 0.1	102 ± 4
	4	100 ± 6	78 ± 8	0.93 ± 0.1	95 ± 6
	8	81 ± 10	65 ± 10	1.0 ± 0.1	77 ± 12
	16	63 ± 8	45 ± 10	1.0 ± 0.1	62 ± 8
	32	44 ± 4	29 ± 5	1.0 ± 0.1	42 ± 4
Alegria	1	140 ± 10	120 ± 9	1.0 ± 0.1	131 ± 12
	2	120 ± 9	100 ± 10	1.1 ± 0.1	112 ± 7
	4	110 ± 8	79 ± 20	1.1 ± 0.1	104 ± 6
	8	88 ± 7	53 ± 8	1.1 ± 0.1	87 ± 6
	16	69 ± 10	39 ± 5	0.93 ± 0.1	71 ± 10
	32	46 ± 2	27 ± 1	0.99 ± 0.1	47 ± 3
Epure	1	150 ± 7	120 ± 10	1.0 ± 0.1	138 ± 6
	2	130 ± 4	110 ± 4	1.1 ± 0.1	110 ± 4
	4	90 ± 4	90 ± 8	1.1 ± 0.1	106 ± 2
	8	98 ± 4	66 ± 10	1.0 ± 0.1	95 ± 2
	16	76 ± 3	43 ± 2	0.97 ± 0.1	77 ± 3
	32	55 ± 2	30 ± 2	1.1 ± 0.2	56 ± 2
Festivo	1	150 ± 20	120 ± 10	1.0 ± 0.1	136 ± 16
	2	120 ± 4	97 ± 10	1.1 ± 0.1	111 ± 3
	4	110 ± 4	75 ± 10	1.2 ± 0.2	102 ± 2
	8	92 ± 5	51 ± 7	1.1 ± 0.1	91 ± 4
	16	77 ± 8	43 ± 4	1.1 ± 0.2	77 ± 7
	32	54 ± 7	30 ± 3	1.0 ± 0.1	54 ± 7
Lyside	1	150 ± 6	130 ± 10	0.98 ± 0.1	140 ± 5
	2	120 ± 5	110 ± 6	1.0 ± 0.1	110 ± 4
	4	110 ± 6	86 ± 20	1.1 ± 0.1	100 ± 4
	8	95 ± 3	60 ± 9	1.1 ± 0.2	92 ± 3
	16	75 ± 2	50 ± 9	0.94 ± 0.1	73 ± 2
	32	51 ± 2	31 ± 1	1.0 ± 0.1	51 ± 2
V316OL	1	140 ± 10	120 ± 10	1.0 ± 0.1	130 ± 10
	2	120 ± 4	110 ± 4	1.1 ± 0.1	110 ± 3
	4	110 ± 5	91 ± 4	1.1 ± 0.1	98 ± 4
	8	91 ± 6	68 ± 10	1.0 ± 0.1	87 ± 5
	16	70 ± 2	49 ± 10	0.97 ± 0.1	68 ± 3
	32	50 ± 3	30 ± 1	1.0 ± 0.1	49 ± 4

Volume weighted mean (*d*_43_), surface weighted mean (*d*_32_).

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
