# Peer review of "Protein Recovery from Rapeseed Press Cake: Varietal and Processing Condition Effects on Yield, Emulsifying Capacity and Antioxidant Activity of the Protein Rich Extract"

_foods, 2019, doi:10.3390/foods8120627_

Round 1
Reviewer 1 Report
Manuscript ID: foods-640572
Title: Protein recovery yield and functional properties of proteins extracted from five different varieties of cold-pressed rapeseed press cake
Authors: Karolina Östbring, Cecilia Tullberg, Emma Malmqvist, Marilyn Rayner
Overview and genral recomendation:
In the manuscript the possibilities of use rapeseed protein, extracted from cold-pressed rapeseed press cake, as multifunctional ingredient with emulsifying capacity and delaying effect on oxidation were considered. The aim of the article was to investigate protein recovery yield, emulsifying properties and oxidative stability of five industrially cold-pressed rapeseed cultivars (Alegria, Epure, Festivo, Lyside, V316OL) and mixed blends. The study included also the protein recovery from rapeseed press cake by modified Wijesundera’s method. The major finding was that the recover rapeseeds protein might be used as a natural, plant origin ingredient with good emulsifying capacity and antioxidant properties comparable with Bovine Serum Albumin (BSA).
Indisputably the subject of the manuscript is very interesting. It describes the possibility of waste material management which is press cake obtained after pressing or extraction of rapeseed oil. Secondly according to newest, highly growing consumer trends, which are vegan food, the plant derived protein, that has emulsifying properties, might be an alternative for animal origin emulsifiers that are widely used in food products. Below I give details about my concerns.
Major comments:
I am concerned about the statistical analysis that was carried out in the manuscript. First of all, the way of presenting statistics on the charts is incomprehensible to me (Figures 2 and 4) - in my opinion, the means marked with letters should be given in ascending order, otherwise it is difficult to compare them with each other (e.g. af). Secondly, the statistics for the values presented in Tables 1, 2a, 2b, figures 3, 7 were not calculated, which makes it impossible to draw appropriate conclusions, as well the results be interpreted appropriately. The proposed statistical analysis – Mann Withney U-test is not one which is typically used in such experiments – in field of food technology it is rather One Way Anova – Tuckey test used. It would also by good to give the name of a program in which this analysis was calculated. I am also interested in explanation why some of determinations are done for two variants of the extraction process of rapeseed proteins – with and without heat (Figure 2, table 1, figure 4, table 2a and 2b, figure 5) and some are made only for one (Figure 6, 7) and it is not specifically explained which variants was used?
Minor comments:
Lines 32-34 refer to limited utilisation of rapeseeds due to the content of glucosinolates which are limited in seeds - Canola rapeseed (Brassica napus and Brassica campestris/rapa) which was bred through standard plant breeding techniques have low levels of erucic acid (< 2%) in the oil portion and low levels of glucosinolates (< 30 μmol/g) in the press cake. The level of glucosinolates in canola has continued to decrease in recent years due to selection pressure by canola plant breeders and is no longer a concern (canolacouncil.org). Line 52 there is an information that chocolate is in emulsion form, which is not true, although in its production the emulsifier as lecithin is being used, but it has different function - it reduces viscosity, replaces expensive ingredients such as cocoa butter, improves the flow properties of chocolate, and can improve the shelf life. Lines 54-55, 151 and in whole text of the manuscript – there is a point that emulsifiers are used to stabilize emulsions, but this is not a reason why words emulsifier and stabilizer are used interchangeably – not every stabilizer works as emulsifier? Line 62 authors suggest that “rapeseed protein might be an alternative emulsifier that have a reduced risk of food allergy” – it should be verified because rapeseed belongs to the same botanical family like mustard seeds, which are on the list of allergens. Line 113-115 – Figure 1 – the heating process should be one of options – from the schematic illustration it can be seen that supernatant that is not heated has not get the pH adjustment – is it correct? In line 122 is an information that after colleting the supernatant the pH was adjusted and after this it was divided into to two groups – one which was heated and other cold processed? Lines 122, 148 is word “incubation” (in temperature 4°C) used properly? Lines 142 and 157 – Please give some literature source base on which were those method modeled on? Those are not standard methods or at least not very common ones. It might give the explanation why emulsions were prepared for every determination in different way, with different ingredients – why for preparation of emulsion there is a different dispersed phase than in stability? Line 150 – the title of the determination is too general considering that only one emulsion properties is being described - particle size distribution, it might be more precise – “2.5 Particle size distribution of emulsions”. There is also another possibility to connect determinations form point 2.5 and 2.6 as they both describe the analysis of emulsions even using the same analyser. Line 159 – why BSA was used? The lecithin is a very popular emulsifier which is use commonly in food product, but BSA is not, there are other proteins like milk or soy more commonly used? Line 212 – “although not significant” - data presented in Figure 2, accompanied by statistical groups indicate that, there are differences between protein recovery yield. Line 220 – Figure 2 is low quality, it is fuzzy Table 1 – all values and standard deviations in columns should be given with the same accuracy, and with dry matter results sometimes are given as ± 0.1 and sometimes ± 1. There is an extra “(“ in mass sedimentation column. Lines 230 and 235 – how literature sources should be cited with year or without year? Line 282 – Table 2a and line 285 table 2b contains data (like D 32, and Span and Mode) that were not mentioned in the text, are they not important? Are the data form table 2a and 2b also presented on figure 3 and also on figure 8 so they are in triplicate? It is not necessary and diagrams from picture 3 are not easy to read, so it would be better to use only table as a source of information. Line 304 – lack of µ after 10 it should be µm. If it is the description for diagram 5b than the large peak is not around 40 µm but rather around 100 µm? Line 332 – it should be explained with literature data how plant breeding efforts do change protein fractions. Line 337 – divide those determinants as two different features – into two chapters, as they are completely different – emulsion stability as physical attribute and oxidative stability as chemical antioxidative ability. Line 354 – The extent for the lecithin with added tocopherol stabilisation is only for 8th day, at the and of experiment (32nd day) all emulsions were at the same level, although there is no statistic data to confirm this observation. Line 424 – it is hard to tell due to statistic presentation which gives no opportunity to compare the means Lines 425-426 - Epure and Festivo had the highest capacity to delay oxidation comparable with BSA, where it can be seen, there is no statistical analysis and from figure 7b is not that visible.
Author Response
Response to Reviewer #1
We have done our best to make the suggested changes and hope you find them satisfactory in the revised manuscript. Please find our responses to the comments below. In the revised manuscript new and significantly modified sections are indicated in yellow.
Statistics. Thanks for the advice to use one-way Anova with the Tukey test instead. We have re-calculated the statistics including the data presented in table 1.
Heat treatment. We compare the anti-oxidative effect with BSA and soy lecithin which all are exposed to heat treatment in the production process. Therefore, we chose the heat-treated set to get a better comparison with the commercial controls. Also, there were only room for a limited number of samples within the experimental framework.
Comment 1. Lines 32-34 refer to limited utilisation of rapeseeds due to the content of glucosinolates which are limited in seeds - Canola rapeseed (Brassica napus and Brassica campestris/rapa) which was bred through standard plant breeding techniques have low levels of erucic acid (< 2%) in the oil portion and low levels of glucosinolates (< 30 μmol/g) in the press cake. The level of glucosinolates in canola has continued to decrease in recent years due to selection pressure by canola plant breeders and is no longer a concern (canolacouncil.org).
Response: Glucosinolate is water soluble and in oil pressing process, the glucosinolate is enriched in press cake. Therefore, it is higher concentrations of glucosinolate in the presscake compared to the rapeseeds. Although the starting concentrations have been dramatically reduced through plant breeding, there is still concerns regarding the protein part and glucosinolates. In another study we are tracing the glucosinolate concentrations throughout the protein upcycling process, but that is outside the scope for this study.
Comment 2. Line 52 there is an information that chocolate is in emulsion form, which is not true, although in its production the emulsifier as lecithin is being used, but it has different function - it reduces viscosity, replaces expensive ingredients such as cocoa butter, improves the flow properties of chocolate, and can improve the shelf life.
Response: We agree and the chocolate has been deleted.
Comment 3. Lines 54-55, 151 and in whole text of the manuscript – there is a point that emulsifiers are used to stabilize emulsions, but this is not a reason why words emulsifier and stabilizer are used interchangeably – not every stabilizer works as emulsifier?
Response: We understand the concerns and have changed throughout the manuscript and used emulsifiers in multiple formulations to avoid misunderstanding.
Comment 4. Line 62 authors suggest that “rapeseed protein might be an alternative emulsifier that have a reduced risk of food allergy” – it should be verified because rapeseed belongs to the same botanical family like mustard seeds, which are on the list of allergens.
Response: In Sweden, we have only had 2 cases of allergenicity to rapeseed in the last 30 years and rapeseed. We understand that the situation in other parts of the world is different as has deleted the sentence.
Comment 5. Line 113-115 – Figure 1 – the heating process should be one of options – from the schematic illustration it can be seen that supernatant that is not heated has not get the pH adjustment – is it correct?
Response: Thanks for the comment, Figure 1 have been redrawn to show both options: pH adjustment was applied for both routes (non heat/heat treatment).
Comment 6. In line 122 is an information that after colleting the supernatant the pH was adjusted and after this it was divided into to two groups – one which was heated and other cold processed?
Response: Yes, that is correct.
Comment 7: Lines 122, 148 is word “incubation” (in temperature 4°C) used properly?
Response: We changed to “mixed for 3 hours” and “stored for 1 hour” (p 4, line 119 and p 4 line 147)
Comment 8. Lines 142 and 157 – Please give some literature source base on which were those method modeled on? Those are not standard methods or at least not very common ones. It might give the explanation why emulsions were prepared for every determination in different way, with different ingredients – why for preparation of emulsion there is a different dispersed phase than in stability?
Response: Other researchers calculate the recovering yield differently. Klockeman and Toledo (1997) is a highly citated source in the rapeseed protein field and they report an impressive yield of 99% but in their opinion they have three products: the sediment from the first centrifugation, the supernatant from the second centrifugation as well as the sediment from the second centrifugation. The last percent that is missing from 100% must be only loss within apparatus. Since we want as high protein concentration as possible we identify only the last sediment as product and want to take all losses into account. In our projects we are trying to find applications also for the proteins in the supernatant (beverage applications for example) but that is not the scope for this study.
In the emulsion part we use Miglyol 812 as oil phase. Miglyol is a stable oil not prone to oxidation. The oil is completely liquid and the composition is also always the same independent of batches. By using this oil it is possible to exclude variability in the oil as a parameter in the emulsion trials. We have used Miglyol to characterize plant based proteins as emulsifiers in other work and can therefore compare results with our previous work (eg. Östbring et al 2015, Heat-induced aggregation of thylakoid membranes affect their interfacial properties, Food Funct 6: 1310-1318). In the oxidation trials, on the other hand we must have an oil with the capacity to oxidize. Preferrably an oil with rapid oxidation rate so we could follow the rate over a reasonable amount of time. We tried different oils on the market but in Sweden it was impossible to find an oil that was not oxidized in the bottle at the purchase. Therefore, we got help from an oil pressing company (Gunnarshögs Gård) and pressed flaxseed oil in the morning and produced the emulsions for the oxidation trial the same day.
Comment 9. Line 150 – the title of the determination is too general considering that only one emulsion properties is being described - particle size distribution, it might be more precise – “2.5 Particle size distribution of emulsions”. There is also another possibility to connect determinations form point 2.5 and 2.6 as they both describe the analysis of emulsions even using the same analyser.
Response: We agree and have combined the former sections 2.5 and 2.6. (p 4, line 149)
Comment 10. Line 159 – why BSA was used? The lecithin is a very popular emulsifier which is use commonly in food product, but BSA is not, there are other proteins like milk or soy more commonly used?
Response: We wanted both lecithin and a protein as control. We tried with casein but it was difficult to disperse so we went for BSA as a representative for the animal proteins instead.
Comment 11. Line 212 – “although not significant” - data presented in Figure 2, accompanied by statistical groups indicate that, there are differences between protein recovery yield.
Response: From the new statistical analysis there were no difference in protein recovery yield between samples that had underwent heat treatment during extraction and those that had not. For the revised version, we chose to present the protein recovery yield in Table 1 instead of a figure.
Comment 12. Line 220 – Figure 2 is low quality, it is fuzzy
Response: We agree. Data presented in former Fig 2 has now been added to Table 1 instead. We submitted a high resolution version of all figures as well and will ask the editors to take that version instead in case of fussiness.
Comment 13. Table 1 – all values and standard deviations in columns should be given with the same accuracy, and with dry matter results sometimes are given as ± 0.1 and sometimes ± 1. There is an extra “(“ in mass sedimentation column.
Response: We have changed so all columns have the same accuracy. The standard deviations are given with equal amount of significant digits (i.e one significant digit).
Comment 14. Lines 230 and 235 – how literature sources should be cited with year or without year?
Response: The year have been deleted.
Comment 15. Line 282 – Table 2a and line 285 table 2b contains data (like D 32, and Span and Mode) that were not mentioned in the text, are they not important?
Response: The parameters are explained in the method section (p 4, line 155-159) and a comment has been added on d43, d32 and mode (p 7 lines 273-276).
Comment 16. Are the data form table 2a and 2b also presented on figure 3 and also on figure 8 so they are in triplicate? It is not necessary and diagrams from picture 3 are not easy to read, so it would be better to use only table as a source of information.
Response: We agree and have deleted Fig 3. There are no Fig 8 and the data in Fig 7 is not included in the table.
Comment 17. Line 304 – lack of µ after 10 it should be µm.
Response: Thanks, the µ has been added (p 10 line 302).
Comment 18: If it is the description for diagram 5b than the large peak is not around 40 µm but rather around 100 µm?
Response: There is a (very small) peak around 10µ in both 5a and 5b. We have rephrased to include also the peak around 100 µm in the figure (p 10, lines 301-303). The figure number are not the same in the revised manuscript. The new figure number is Figure 3.
Comment 19. Line 332 – it should be explained with literature data how plant breeding efforts do change protein fractions.
Response: Thanks for the comment. The protein fractions (i.e the proportion between the different proteins) might be hard to change. I was thinking of total amount of protein but that is not applicable here and we have deleted the sentences. Thanks.
Comment 20. Line 337 – divide those determinants as two different features – into two chapters, as they are completely different – emulsion stability as physical attribute and oxidative stability as chemical antioxidative ability
Response: We agree and a new sub heading has been added (p 11, line 348)
Comment 21. Line 354 – The extent for the lecithin with added tocopherol stabilisation is only for 8th day, at the and of experiment (32nd day) all emulsions were at the same level, although there is no statistic data to confirm this observation.
Response: The other reviewer suggested to delete the samples with alpha tocopherol, because no citric acid was used. We were not aware of this practice and have agreed to delete the samples from the graph and in the manuscript.
Comment 22. Line 424 – it is hard to tell due to statistic presentation which gives no opportunity to compare the means.
Response: We have performed a one-way ANOVA with Tukeys test and the significant differences are now indicated the text (eg. Fig 2).
Comment 23. Lines 425-426 - Epure and Festivo had the highest capacity to delay oxidation comparable with BSA, where it can be seen, there is no statistical analysis and from figure 7b is not that visible.
Response: We have performed a one-way ANOVA with Tukeys test and the significant differences are now indicated in Fig 5b.

Reviewer 2 Report
The manuscript entitled “Protein recovery yield and functional properties of proteins extracted from five different varieties of cold-pressed rapeseed press cake” dials with the extraction and valorization of proteins from a rapeseed oil industry byproduct. First of all the title, in my opinion, is not appropriate. In fact, authors evaluated the emulsifying capacity and the antioxidant activity of the extracted proteins. These results are not sufficient to verify if the extracted proteins could be considered functional food. I suggest the following title “Protein recovery from rapeseed press cake: varietal and processing condition effects on yield, emulsifying capacity and antioxidant activity of the protein rich extract”. As the keywords regard, I suggest to the authors to reduce the use of composed words and to add “by-product”. There is an important weakness in the experimental method that is related to varietal comparison. As the authors surely know, there are important effects of agronomical and pedoclimatic conditions on the rapeseed composition. These effects are responsible of important variability that should be considered in an experiment aimed to compare varieties. For this reason the presented results could be considered only preliminary.
ABSTRACT
I suggest to the authors to write again the abstract, improving the English stile and clarity. For instance, “…all varieties investigated had similar emulsifying capacity…” the subject are not the varieties but the proteins extracted from each varieties. The same for the antioxidant activity. L18-20 Phrase not clear and speculative…
INTRODUCTION
L26 “…globally” add cultivated
L68 symbol of linolenic acid missed
MATERIALS AND METHODS
Fig1. Considering what the authors reported in the text, after the first centrifugation citric acid solution was added to all samples. Then, two aliquots were prepared, one of which was heated at 80°C for 3 min and the other not heated, before the following centrifugation step. So, the diagram must be corrected.
The evaluation of the emulsifying capacity of a surfactant should consider the determination of the “Critical Micelle Concentration” (CMC) that is a very important characteristic.
In the paragraph analysis of emulsions please explain the meaning of the terms d32 and d43 and the significance of this parameter in the particle size description.
The decision to use a-tocopherol as antioxidant was incorrect. It is well known that a-tocopherol in particular, could act as pro-oxidant (Cillard, Cillard, & Cormier, 1980; Cillard, Cillard, Cormier, & Girre, 1980). For this reason in food industry a-tocopherols is always used together with ascorbic acid. For this reason the obtained results which highlight an increasing oxidation rate of the mixture containing a-tocopherol is expected.
Please explain the reason for choosing the Bovine Serum Albumin as reference for antioxidant activity in the emulsions.
RESULTS AND DISCUSSION
Fig.2 The resolution must be improved; moreover, it is very difficult to understand the significative comparisons. The letter “b” on the Lyside heated shows a significative difference in respect to Festivo no heated and no significative difference in comparison to Mixed heated…Please check all figures for significance letters.
Table 1. Report in the caption the meaning of DM (dry matter) and RPC (Recovered Protein Concentration??)
Figure 3. Lines belonging to different varieties are indiscernible.
L304 Symbol missed for 10 micrometer.
EMULSION STABILITY AND LIPID OXIDATION
Considering the reported results, I suggest to the author to delete in the manuscript al data related to the use of BSA and a-tocopherol leaving only the soy lecithin test as reference. As above reported, the use of these standards was inappropriate for the scope.
CONCLUSIONS
The conclusions must be write again considering that the reported results are only preliminary and must be verified by a minimum experimentation of three years .
In the all, the paper could be published only after major revisions of the content and the English form.
Author Response
Response to Reviewer #2
Thanks for the valuable comments. We have made changes according to the suggestions and new or significantly modified sections are indicated in yellow. We feel that the comments have improved the manuscript significantly.
Title. Thanks for the suggestion of a new (and improved) title. We agree and have changed (p 1, lines 1-5).
Varietal comparison. Thanks for the comment. The rapeseed varieties were cultivated and harvested the same season at the same farm in Southern Sweden. We have added this information in the Method section (p2, line 90-92). We agree that this study is only preliminary and has added a comment in the text as well as in the conclusion section (p 5, lines 208-211, p 12. Lines 386-388).
Key words. We agree and have changed to “by-product”.
Comment 1. I suggest to the authors to write again the abstract, improving the English stile and clarity. For instance, “…all varieties investigated had similar emulsifying capacity…” the subject are not the varieties but the proteins extracted from each varieties. The same for the antioxidant activity. L18-20 Phrase not clear and speculative…
Response: The abstract has been rewritten (P 1, lines 14-18).
Comment 2. L26 “…globally” add cultivated.
Response: “cultivated” have been added (p1, line 26)
Comment 3. L68 symbol of linolenic acid missed
Response: symbol has been added (P2, line 66).
Comment 4. Fig1. Considering what the authors reported in the text, after the first centrifugation citric acid solution was added to all samples. Then, two aliquots were prepared, one of which was heated at 80°C for 3 min and the other not heated, before the following centrifugation step. So, the diagram must be corrected.
Response: Thanks for the comment, Fig 1 has been redrawn and is now including addition of citric acid in both process lines.
Comment 5. The evaluation of the emulsifying capacity of a surfactant should consider the determination of the “Critical Micelle Concentration” (CMC) that is a very important characteristic.
Response: Since rapeseed protein is a protein, there is no CMC. Proteins do not form micelles and the amount of emulsifiers is limited to as much as can fit on the surface.
Comment 6. In the paragraph analysis of emulsions please explain the meaning of the terms d32 and d43 and the significance of this parameter in the particle size description.
Response: A comment on particle size distributions and particle diameters (d32 and d43) has been added (p4, lines 155-158).
Comment 7. The decision to use a-tocopherol as antioxidant was incorrect. It is well known that a-tocopherol in particular, could act as pro-oxidant (Cillard, Cillard, & Cormier, 1980; Cillard, Cillard, Cormier, & Girre, 1980). For this reason in food industry a-tocopherols is always used together with ascorbic acid. For this reason the obtained results which highlight an increasing oxidation rate of the mixture containing a-tocopherol is expected.
Response: According to the comment further down, the a-tocopherol data have been deleted.
Comment 8. Please explain the reason for choosing the Bovine Serum Albumin as reference for antioxidant activity in the emulsions.
Response: We produced emulsions with rapeseed proteins as emulsifiers and wanted controls from both plant and animal origin. We also tried with casein but it was very difficult to disperse so we went for BSA instead as the animal protein alternative. Oleosin and BSA are in the same size range and is much larger compared to lecithin. The hypothesis was that the proteins will cover the surface to a higher degree compared to the small lecithin molecule and thereby delay oxidation to a greater extent.
Comment 9. Fig.2 The resolution must be improved; moreover, it is very difficult to understand the significative comparisons. The letter “b” on the Lyside heated shows a significative difference in respect to Festivo no heated and no significative difference in comparison to Mixed heated…Please check all figures for significance letters.
Response: According to a suggestion from the other reviewer, the statistics have been recalculated using one-way ANOVA and the new significant letters have been double checked.
Comment 10. Table 1. Report in the caption the meaning of DM (dry matter) and RPC (Recovered Protein Concentration??)
Response: RSPC has been explained in the table caption (p6, line 223). Also, Fig 2 has been deleted and the data is now presented table 1 as a new column. Dry basis has been written out to make it easier to understand the table.
Comment 11. Figure 3. Lines belonging to different varieties are indiscernible.
Response: According to comments from the other reviewer we have deleted Fig 3 and refer to the d43 data in Table 2 instead.
Comment 12. L304 Symbol missed for 10 micrometer.
Response: Thanks, the symbol has been added (p 10, line 302).
Comment 13. Considering the reported results, I suggest to the author to delete in the manuscript al data related to the use of BSA and a-tocopherol leaving only the soy lecithin test as reference. As above reported, the use of these standards was inappropriate for the scope.
Response: We agree to delete samples with alpha tocopherol but we want to keep the BSA. It is not reasonable to only compare lecithin and rapeseed protein. We feel that the comparison with BSA is needed to get the comparison with another protein as well.
Comment 14. The conclusions must be write again considering that the reported results are only preliminary and must be verified by a minimum experimentation of three years.
Reponse: A comment about that the study should be considered preliminary and that more robust studies are needed has been added to the conclusion section (p 12, lines 386-388).

Round 2
Reviewer 1 Report
Manuscript ID: foods-640572 (review of the revised version)
Title: Protein recovery yield and functional properties of proteins extracted from five different varieties of cold-pressed rapeseed press cake
Authors: Karolina Östbring *, Cecilia Tullberg, Emma Malmqvist, Marilyn Rayner
Overview and genral recomendation:
Although authors made an efforted and corrected most of my concerns in the manuscript there are still some parts that were left without correction.
Major comments:
The values presented in Tables 2a, 2b, are still not statistically calculated, which makes it impossible to draw appropriate conclusions, as well the results be interpreted appropriately.
Although there is (in comments to the Reviewer 1) an explanation why some of determinations are done for two variants of the extraction process of rapeseed proteins – with and without heat and others are made only for one. I think that this also should be explained in the methodology of the experiment? And the explanation that there was not enough space/room in my opinion is insufficient, there must have been a good reason for selecting some variants and rejecting others?
Minor comments:
Comment 15. Line 282 – Table 2a and line 285 table 2b contains data (like D 32, and Span and Mode) that were not mentioned in the text, are they not important?
Authors’ Response: The parameters are explained in the method section (p 4, line 155-159) and a comment has been added on d43, d32 and mode (p 7 lines 273-276).
Reviewer’s Response: In new version of the article there are not lines 273-276, or maybe there is something wrong with my version od PDF? On p 7 (and in whole manuscript) there is no reference in the text about d32 parameter, it is just presented on the graphs and in the table.
Comment 23. Lines 425-426 - Epure and Festivo had the highest capacity to delay oxidation comparable with BSA, where it can be seen, there is no statistical analysis and from figure 7b is not that visible.
Response: We have performed a one-way ANOVA with Tukeys test and the significant differences are now indicated in Fig 5b.
Reviewer’s Response: In the conclusion part (line 784) is now better to say, that “all varieties except the mix one(s) had significantly higher capacity to delay oxidation” so generally rapeseed protein are better antioxidants than BSA or lecithin?
Author Response
Dear Reviewer,
Thanks for the comments. We have done our best to make the suggested changes and hope you find them satisfactory in the revised manuscript. Please find our responses to the comments below. In the revised manuscript new and significantly modified sections are indicated in green.
Overview and genral recomendation:
Although authors made an efforted and corrected most of my concerns in the manuscript there are still some parts that were left without correction.
Major comments:
Comment 1: The values presented in Tables 2a, 2b, are still not statistically calculated, which makes it impossible to draw appropriate conclusions, as well the results be interpreted appropriately.
Response: The data in Table 2a and 2b has been statistically analyzed (se comments below for comments on the method). Area under the curve was calculated to reflect the emulsifying capacity over the whole range of protein concentrations used in the emulsion experiments. Data was analysed in SPSS using different methods depending on if the data was normally distributed or not. Comments in the manuscript has been added in the Method section (p5, lines 201-210) and in the Results and discussion section (p7, lines 281-289 and 297-301). For the present paper, the results from Fig 2 is more important since 8 mg protein/ml oil is the critical concentration for this system (explained in p9 lines 297-301). We still want to share emulsion data in table 2 for all protein concentrations since this information gives readers in the emulsion field valuable information over the larger concentration scale. Exactly how the different data points in the table is different from each other are of less importance, so therefore we chose to do area under the curve to get information about the total emulsifying capacity over the whole protein concentration range.
Comment 2: Although there is (in comments to the Reviewer 1) an explanation why some of determinations are done for two variants of the extraction process of rapeseed proteins – with and without heat and others are made only for one. I think that this also should be explained in the methodology of the experiment? And the explanation that there was not enough space/room in my opinion is insufficient, there must have been a good reason for selecting some variants and rejecting others?
Response: We have added a comment in the method section in order to be clear with which process the proteins in the emulsion stability tests and oxidation tests and why we focused on heat-treated rapeseed proteins. P 4, line 164-167 and p 5 line 176-176.
Minor comments:
Comment 15. Line 282 – Table 2a and line 285 table 2b contains data (like D 32, and Span and Mode) that were not mentioned in the text, are they not important?
Authors’ Response: The parameters are explained in the method section (p 4, line 155-159) and a comment has been added on d43, d32 and mode (p 7 lines 273-276).
Reviewer’s Response: In new version of the article there are not lines 273-276, or maybe there is something wrong with my version od PDF? On p 7 (and in whole manuscript) there is no reference in the text about d32 parameter, it is just presented on the graphs and in the table.
Response: Data from table 2 has been statistically evaluated by means of area under the curve and d43, d32, mode and span is now commented ((p7, lines 281-289 and 297-301)
Comment 23. Lines 425-426 - Epure and Festivo had the highest capacity to delay oxidation comparable with BSA, where it can be seen, there is no statistical analysis and from figure 7b is not that visible.
Response: We have performed a one-way ANOVA with Tukeys test and the significant differences are now indicated in Fig 5b.
Reviewer’s Response: In the conclusion part (line 784) is now better to say, that “all varieties except the mix one(s) had significantly higher capacity to delay oxidation” so generally rapeseedprotein are better antioxidants than BSA or lecithin?
Response: After recalculation of the statistics according to the other reviewer, only two varieties (Festivo and Epure) had higher capacity to delay oxidation compared to soy lecithin. We consulted a colleague with experience of non-parametric data in the food area and did a thorough investigation of the data. Some datasets were normal distributed, and some were not (see Method section p5, lines 201-210 for further details). For the normal distributed sets, we used a univariate general lineal model with Tukey’s test and for the non-parametric data sets we used Kruskal Wallis test with Bonferroni-adjusted pairwise comparison. The figures and tables have been double checked to make sure that the new significant letters are correct. The majority of the results were similar to our first version, but some results were not significant anymore, since non-parametric test is more “strict” compared to parametric tests. The manuscript has been updated to reflect the new statistical analysis.
Reviewer 2 Report
I am satisfied of the given answers by the authors and I think that the revised form of the manuscript has significative improvement. Unfortunately I still have concern about the English style which require a deeply revision from a native English speaker. Look at the paragraph "3.2 Emulsifying properties" lines 445-452 of the revised version; the first phrase is a repetition, we are in the results section, delete; samples showed the same trend not the same pattern; the droplet size decreased with the increase of the amount of proteins added in each samples set (this is the sense, I guess); the following phrase (L449-452) is unreadable.
About the statistical method described in the material and methods, there is a problem. Firstly the author stated that they used the Mann Whitney U-test to detect significant differences among samples. The Mann Whitney U-test is a non-parametric test used to compare two populations of data. In the revised form of the manuscript the authors changed the paragraph and indicated the use of ANOVA followed by Tuky test. Then the authors said that when the ANOVA was significant they used the Mann Whitney U-test for multiple comparisons. First of all, the authors have to indicate if they are working with parametric or non-parametric data. As they surely know a parametric variable must fulfil the tests of normal distribution and the homogeneity of variance other than a continuous interval of distribution and must be independent one from each other’s. If the variables measured from the authors are not parametric and the compared treatments are more than two (as in this case) the appropriate test is the Friedman's test. If the variables are parametric and there are multiple comparisons the author have to use the ANOVA followed by the Tukey test also named Tukey's HSD (honestly significant difference). A part this problem, I am satisfied about the answers given by the authors to all my questions, and after a deeply revision of the English style.
Author Response
Dear Reviewer,
Thanks for the valuable comments in the second round. We have done our best to make the suggested changes and hope you find them satisfactory in the revised manuscript. Please find our responses to the comments below. In the revised manuscript new and significantly modified sections are indicated in green.
I am satisfied of the given answers by the authors and I think that the revised form of the manuscript has significative improvement. Unfortunately I still have concern about the English style which require a deeply revision from a native English speaker. Look at the paragraph "3.2 Emulsifying properties" lines 445-452 of the revised version; the first phrase is a repetition, we are in the results section, delete; samples showed the same trend not the same pattern; the droplet size decreased with the increase of the amount of proteins added in each samples set (this is the sense, I guess); the following phrase (L449-452) is unreadable.
Response: The text has been revised by a native English-speaking colleague. If the reviewer feels that the language in the manuscript is not acceptable, we can send it to an external professional. Please let us know.
About the statistical method described in the material and methods, there is a problem. Firstly the author stated that they used the Mann Whitney U-test to detect significant differences among samples. The Mann Whitney U-test is a non-parametric test used to compare two populations of data. In the revised form of the manuscript the authors changed the paragraph and indicated the use of ANOVA followed by Tuky test. Then the authors said that when the ANOVA was significant they used the Mann Whitney U-test for multiple comparisons. First of all, the authors have to indicate if they are working with parametric or non-parametric data. As they surely know a parametric variable must fulfil the tests of normal distribution and the homogeneity of variance other than a continuous interval of distribution and must be independent one from each other’s. If the variables measured from the authors are not parametric and the compared treatments are more than two (as in this case) the appropriate test is the Friedman's test. If the variables are parametric and there are multiple comparisons the author have to use the ANOVA followed by the Tukey test also named Tukey's HSD (honestly significant difference). A part this problem, I am satisfied about the answers given by the authors to all my questions, and after a deeply revision of the English style.
Response: We are very grateful for the comment and have consulted a colleague with experience of non-parametric data sets in the food area. Together we did a thorough investigation of the data. Some datasets were normal distributed, and some were not (see p 5, lines 201-210 for further details). For the normal distributed sets, we used a univariate general lineal model with Tukey’s test and for the non-parametric data sets we used Kruskal Wallis test with Bonferroni-adjusted pairwise comparison. From what we read the Friedmans test is used for paired data but the data in the present study is un-paired. We therefore used the univariate general lineal model with Tukey’s test. The figures and tables have been updated with results from the new statistical analysis and have been double checked to make sure that the new significant letters are correct. Also, the manuscript has been updated with the results from the new statistical evaluation.